# MOLGA: MOLECULAR GRAPH ADAPTATION WITH PRE-TRAINED 2D GRAPH ENCODER

## ABSTRACT

Molecular graph representation learning is widely used in chemical and biomedical research. While pre-trained 2D graph encoders have demonstrated strong performance, they overlook the rich molecular domain knowledge associated with submolecular instances (atoms and bonds). While molecular pre-training approaches incorporate such knowledge into their pre-training objectives, they typically employ designs tailored to a specific type of knowledge, lacking the flexibility to integrate diverse knowledge present in molecules. Hence, reusing widely available and well-validated pre-trained 2D encoders, while incorporating molecular domain knowledge during downstream adaptation, offers a more practical alternative. In this work, we propose MOLGA, which adapts pre-trained 2D graph encoders to downstream molecular applications by flexibly incorporating diverse molecular domain knowledge. First, we propose a molecular alignment strategy that bridge the gap between pre-trained topological representations with domain-knowledge representations. Second, we introduce a conditional adaptation mechanism that generates instance-specific tokens to enable fine-grained integration of molecular domain knowledge for downstream tasks. Finally, we conduct extensive experiments on eleven public datasets, demonstrating the effectiveness of MOLGA. Codes are available at https://anonymous.4open.science/r/MolGa-43F3 for anonymous reviewing.

## 1 INTRODUCTION

Molecular graph representation learning has emerged as a mainstream technique in computational chemical and biomedical research (Gilmer et al., 2017; Rong et al., 2020). Early studies primarily leveraged graph encoders, such as graph neural networks (GNNs) (Kipf & Welling, 2017; Veličković et al., 2018) and transformers (Ying et al., 2021; Yun et al., 2019), to encode 2D topological structures of molecules, where nodes represent atoms and edges represent chemical bonds. However, their effectiveness relies heavily on large amounts of labeled data, which are expensive to obtain through labor-intensive wet-lab experiments or computationally demanding physics-based simulations. To overcome this limitation, graph pre-training methods have emerged as a promising solution. They typically follow a two-stage paradigm: first, the graph encoder learns task-agnostic and intrinsic properties from large-scale unlabeled graphs in a self-supervised manner; then, it is fine-tuned on downstream tasks using limited labeled data (Luong & Singh, 2023; Sypetkowski et al., 2024; Yang et al., 2024; Méndez-Lucio et al., 2024).

Beyond 2D topological structures, molecules contain a variety of molecular domain knowledge that characterizes the chemical and physical properties of submolecular instances (atoms and bonds), as illustrated in Fig. 1(a). For example, 3D conformations (Schütt et al., 2017; Zaidi et al., 2023) capture the spatial arrangement of atoms in three-dimensional space. Chemical bond types describe different bonding relationship between atoms, such as single and double bonds (Liu et al., 2022d; Wang et al., 2022). Atom energy reflects the intrinsic chemical energy associated with each atom or bond (Zou et al., 2023; Wang et al., 2025). Leveraging such rich information, recent studies have extended 2D graph learning methods by integrating molecular domain knowledge into graph encoders to enhance their expressiveness (Zaidi et al., 2023; Wang et al., 2025). Additionally, molecular domain knowledge has been incorporated into pre-training objectives (Liu et al., 2022b; Fang et al., 2022; Yu et al., 2024a), leading to the development of molecular pre-training methods.

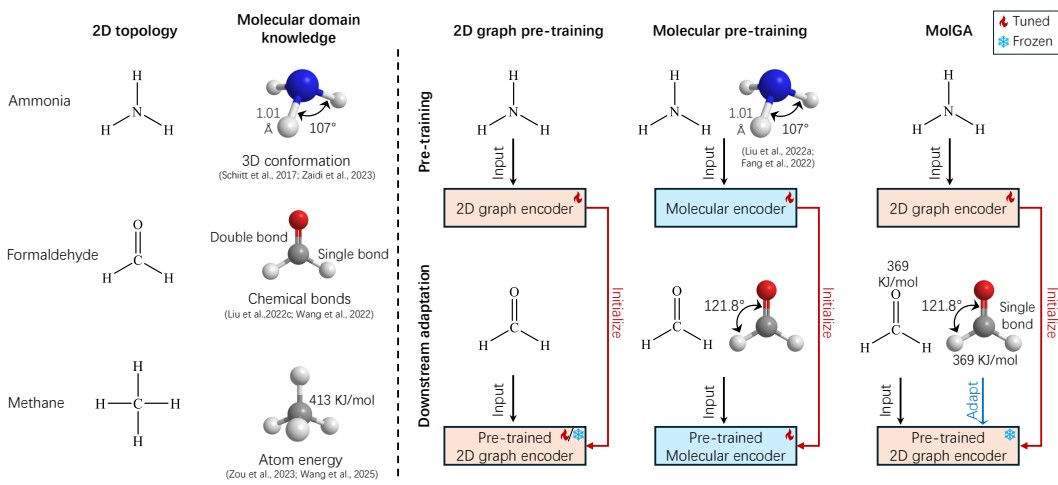

Figure 1: Motivation of MOLGA. (a) 2D topological and molecular domain knowledge in molecules. (b) Comparison of pre-training and downstream adaptation settings in 2D pre-training methods, molecular pre-training approaches and MOLGA.

Current molecular pre-training approaches typically incorporate molecular domain knowledge during the pre-training phase (Liu et al., 2022b; Fang et al., 2022), and the pre-trained model are further fine-tuned for downstream tasks involving the same type of domain knowledge. However, a plethora of well-established pre-trained 2D graph encoders already exist, demonstrating robustness and generalization with extensive empirical validation (Sypetkowski et al., 2024; Yang et al., 2024). Leveraging existing pre-trained 2D graph encoders could significantly reduce costs compared with pre-training a new molecular encoder, which generally involves greater complexity and a larger number of parameters (Schütt et al., 2017; Liu et al., 2022d; Wang et al., 2022). Moreover, existing molecular pre-training methods typically focus on a single type of domain knowledge, thereby overlooking the benefits of incorporating different forms of molecular information (Liu et al., 2023b; Zhou et al., 2023; Kim et al., 2023; Wang et al., 2025). These observations naturally raise a fundamental question: *Given a pre-trained 2D graph encoder, can downstream adaptation flexibly integrate different types of molecular domain knowledge to enhance its performance in molecular applications?* In this work, we propose a **Mol**ecular **G**raph **A**daptation framework, MOLGA, which adapts pre-trained 2D graph encoders to downstream molecular tasks by flexibly incorporating diverse molecular domain knowledge. A comparative overview of the pre-training and downstream adaptation settings across 2D graph pre-training, molecular pre-training, and MOLGA is illustrated in Fig. 1(b). The realization of MOLGA is non-trivial due to two key challenges.

First, *how do we align pre-trained 2D topological representations with molecular domain knowledge for downstream adaptation?* Pre-trained 2D graph encoders typically capture flattened topological patterns, such as atom connectivity and graph motifs. In contrast, molecular domain knowledge differs across submolecular instances (atoms and bonds), with instance-specific chemical or physical characteristics that can vary among similar topological patterns. This discrepancy leads to conflict, rather than synergy, when the two sources of knowledge are naïvely fused in downstream adaptation. In MOLGA, we employ a contrastive alignment strategy that bridges the gap between pre-trained topology and molecular domain knowledge. Specifically, for each atom and bond, we aim to maximize the similarity between its 2D topological representation and its domain-knowledge representation, while minimizing similarity across different instances.

Second, *how can we leverage the aligned knowledge to adapt to fine-grained, instance-specific characteristics for molecular applications?* Existing molecular pre-training approaches typically leverage a one-size-fits-all adaptation (fine-tuning) strategy for downstream tasks (Liu et al., 2022b; Fang et al., 2022; Zong et al., 2024; Wang et al., 2025), disregarding the heterogeneity of submolecular instances. Within a molecule, different atoms and bonds exhibit heterogeneous topological, chemical and physical properties, reflecting their unique roles in the molecule. For example, in formaldehyde ($CH_2O$), the oxygen atom, due to its high electronegativity, renders the C=O bond highly polar-

ized; the carbon atom bears a partial positive charge, making it a favorable site for nucleophilic attack; the C–H bonds are relatively stable, primarily serving to maintain the structural framework of the molecule (Xu et al., 2018). Therefore, fine-tuning all submolecular instances uniformly overlooks such intrinsic variability, limiting the expressiveness and effectiveness of their downstream representations. In MOLGA, we introduce a parameter-efficient, instance-specific adaptation strategy. Inspired by conditional prompting (Zhou et al., 2022; Yu et al., 2025b), MOLGA employs lightweight conditional networks that condition on the aligned instance-level representations to generate a series of instance-specific tokens. The atom-specific tokens are used to adjust atom features before feeding them into the frozen pre-trained graph encoder, while bond-specific tokens are injected into the message passing process. By updating only the conditional networks while keeping the encoder frozen, MOLGA enables fine-grained yet lightweight adaptation to downstream tasks.

In summary, the contributions of this work are threefold. (1) We propose a contrastive alignment strategy to bridge the representational gap between pre-trained 2D topological knowledge and downstream molecular domain knowledge. (2) Realizing that various submolecular instances exhibit distinct topological, chemical and physical characteristics, we introduce an instance-specific downstream adaptation mechanism using lightweight conditional networks. (3) We conduct extensive experiments on a wide array of benchmarks, demonstrating that MOLGA consistently outperforms state-of-the-art methods across various molecular applications.

## 2 RELATED WORK

We briefly review related work on molecular graph representation and pre-training.

**Molecular graph representation learning.** To model the 2D topological structure in molecules, GNNs (Kipf & Welling, 2017; Veličković et al., 2018) and transformers Yun et al. (2019); Ying et al. (2021) have become mainstream techniques. While GNNs aggregate information from local neighborhoods through message passing, Transformers leverage attention to capture interactions between nodes across the graph. However, molecules are also associated with rich physical and chemical domain knowledge, such as 3D conformations and chemical bond types. Hence, recent studies have further incorporated molecular domain knowledge (Schütt et al., 2017; Liu et al., 2022d; Wang et al., 2022), embedding such information directly into node representations. Despite their success, both 2D graph and molecular encoders require substantial labeled data to train for each new task and dataset, limiting their generalizability and scalability.

**Molecular graph pre-training.** To address the limitations of supervised methods, graph pre-training techniques have been developed to capture task-agnostic structural patterns in a self-supervised manner (Kipf & Welling, 2016; Hu* et al., 2020; You et al., 2020; Hu et al., 2020; Yu et al., 2024b). These methods typically pre-train a 2D graph encoder to learn inherent properties from unlabeled graphs and then adapt (fine-tune) the encoder on downstream tasks (Luong & Singh, 2023; Sypetkowski et al., 2024; Yang et al., 2024; Méndez-Lucio et al., 2024). While 2D graph pre-training approaches focus on topological structures, molecular-specific pre-training methods have been proposed to capture rich molecular domain knowledge. Some approaches focus on denoising or reconstructing the 3D geometry of molecules during pre-training (Liu et al., 2023b; Zhou et al., 2023; Kim et al., 2023; Wang et al., 2025), while others incorporate both 2D topological structures and molecular knowledge, including stereochemistry and atom-level interactions, into their pre-training objectives (Stärk et al., 2022; Liu et al., 2022a; 2023a; Chen et al., 2023; Zong et al., 2024). However, existing molecular pre-training methods typically adopt a one-size-fits-all downstream adaptation for all instances, overlooking the distinct chemical and physical characteristics exhibited by different instances. Moreover, these methods require pre-training a molecular graph encoder with a significantly larger number of parameters compared to 2D graph encoders.

## 3 PRELIMINARIES

In this section, we introduce related preliminaries and the scope of our work.

**Molecular graph.** A 2D graph is defined as $G = (\mathcal{V}, \mathcal{E})$, where $\mathcal{V}$ and $\mathcal{E}$ denotes the set of nodes (atoms) and edges (bonds), respectively. A molecular graph is defined as $G = (\mathcal{V}, \mathcal{E}, \mathcal{M})$, which

further includes a set of molecular domain knowledge $\mathcal{M} = \{M_1, M_2, \ldots, M_K\}$. Each $M_k \in \mathcal{M}$ represent a type of domain knowledge on atoms or bonds. Moreover, each atom $v_i \in \mathcal{V}$ is associated with a feature vector $\mathbf{x}_i \in \mathbb{R}^d$, and the entire atom feature matrix is denoted as $\mathbf{X} \in \mathbb{R}^{|\mathcal{V}| \times d}$. We write a collection of molecular graphs as $\mathcal{G} = \{G_1, G_2, \ldots, G_N\}$.

**2D Graph encoder.** A typical implementation utilizes GNNs, which update each node's embedding by iteratively aggregating information from its local neighborhood. Let $\mathbf{H}_{\mathcal{V}}^l \in \mathbb{R}^{|\mathcal{V}| \times d}$ be the node embedding matrix at the $l$-th layer, where each row $\mathbf{h}_i^l$ represents the embedding of node $v_i$. In layer $l$, the forward propagation is given by $\mathbf{h}_v^l = \text{MP}(\mathbf{h}_v^{l-1}, \{\mathbf{h}_u^{l-1} : u \in \mathcal{N}_v\}; \theta^l)$, where $\mathcal{N}_v$ is the neighboring atoms of $v$, $\text{MP}(\cdot)$ is the message-passing function and $\theta^l$ denotes the learnable parameters of the $l$-th layer. In the first layer, the the input feature vector $\mathbf{x}_v$ serves as the initial embedding $\mathbf{h}_v^0$, and the final output of the GNN after $L$ layers is denoted as $\mathbf{h}_v$, which is a row of the node embedding matrix $\mathbf{H}_{\mathcal{V}}$. We denote the overall encoding process over $L$ layers as

$$\mathbf{H}_{\mathcal{V}} = \text{GE}(\mathbf{X}, G; \Theta), \tag{1}$$

where $\text{GE}(\cdot)$ denotes the 2D graph encoder, and $\Theta = \{\theta^1, \theta^2, \ldots, \theta^L\}$ represents the set of trainable parameters across all layers. For the edge embedding matrix $\mathbf{H}_{\mathcal{E}}$, each row $\mathbf{h}_{(u,v) \in \mathcal{E}}$ is computed as the concatenation of the embeddings of the connected nodes, $\mathbf{h}_{(u,v)} = \text{Concat}(\mathbf{h}_u, \mathbf{h}_v)$. The pre-trained $\text{GE}$ with learned weights $\Theta_0$ can be fine-tuned for downstream tasks.

# 4 PROPOSED APPROACH

In this section, we introduce MOLGA, starting with a high-level overview of the framework, and then detail its core components.

## 4.1 OVERALL FRAMEWORK

We illustrate the overall framework of MOLGA in Fig. 2(b), which comprises two core components: molecular alignment and molecular adaptation. Given a pre-trained 2D graph encoder in Fig. 2(a), MOLGA first aligns the pre-trained topological representations with downstream molecular domain knowledge, as shown in Fig. 2(c). For each type of molecular domain knowledge, MOLGA trains a projector to map its representation into the pre-trained embedding space, and further employs a contrastive strategy to align the submolecular instances across their topological and molecular knowledge-based representations. Next, conditioned on the aligned representations, we employ conditional networks to generate instance-specific tokens, as depicted in Fig. 2(d). These tokens incorporate fine-grained, instance-specific molecular domain knowledge into the adaptation process without updating the pre-trained 2D graph encoder, either by modifying the input features to the frozen encoder or by injecting them into the message-passing process.

## 4.2 MOLECULAR ALIGNMENT

**2D topological encoding.** We first feed the 2D version of the downstream molecular graph $G$ into the pre-trained 2D graph encoder $\text{GE}$ to obtain the atom embedding matrix $\mathbf{H}_{\mathcal{V}}$ and the bond embedding matrix $\mathbf{H}_{\mathcal{E}}$ as introduced in Sect. 3.

**Molecular domain knowledge extraction.** For the $k$-th type of domain knowledge $M_k \in \mathcal{M}$, we employ a rule-based extractor $\text{Extr}_k(\cdot)$, following prior work (Wang et al., 2022), to obtain the initial representations of $M_k$, as follows.

$$\mathbf{M}_k = \text{Extr}_k(M_k). \tag{2}$$

The rule-based extractors are non-parametric and their details are presented in Appendix D.

**Molecular domain knowledge projection.** Next, we map the initial representations of each type of molecular domain knowledge into the pre-trained atom embedding space by learnable projectors:

$$\tilde{\mathbf{M}}_k = f_k(\mathbf{M}_k; \phi_k), \tag{3}$$

where $f_k(\cdot)$ denotes the projector for the $k$-th type of knowledge, parameterized by learnable weights $\phi_k \in \Phi$. The output $\tilde{\mathbf{M}}_k$ shares the same dimension as its corresponding atom embedding $\mathbf{H}_{\mathcal{V}}$ or

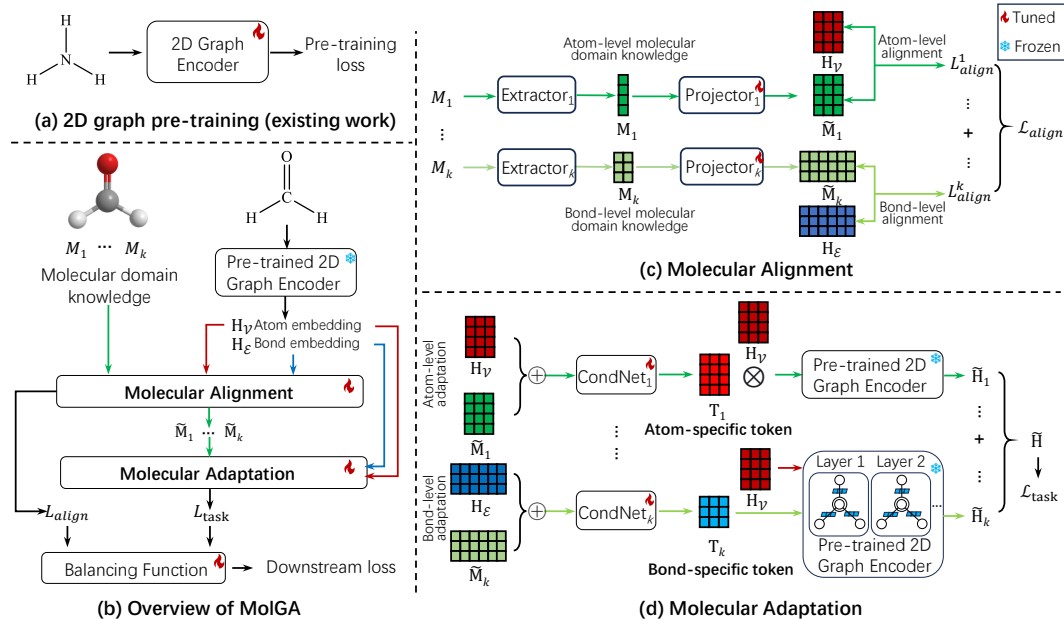

Figure 2: Overall framework of MOLGA. Building upon an existing pre-trained 2D encoder, MOLGA performs molecular alignment and adaptation for downstream tasks.

edge embedding $\mathbf{H}_{\mathcal{E}}$, depending on whether $M_k$ is atom-level or bond-level knowledge. Let $\tilde{\mathbf{m}}_{\mathbf{i},\mathbf{k}}$ be the $i$-th row in $\tilde{\mathbf{M}}_k$, corresponding to the $k$-th type information for the $i$-th atom or bond.

**Contrastive alignment.** While the projection ensures that $\tilde{\mathbf{M}}_k$ and the instance representations ($\mathbf{H}_{\mathcal{V}}$ or $\mathbf{H}_{\mathcal{E}}$) share the same embedding dimension, they are not inherently aligned. Thus, we employ a contrastive alignment strategy to explicitly bridge the representations derived from molecular domain knowledge and topology. Ideally, the 2D topological representation of an instance should implicitly reflect its chemical and physical properties (Pairault et al., 2023; Koppe et al., 2025). Hence, similar in spirit to multimodal alignment (Radford et al., 2021), we adopt a contrastive learning strategy to align the two sources of representation for each instance. Specifically, we pull the 2D and domain-knowledge representations of the same instance closer in the latent space, while pushing apart those from different instances. Formally, given a molecular graph $G$, the contrastive alignment loss for the $k$-th type of domain knowledge ($M_k$) is defined as:

$$L_{\text{align}}^k(\phi_k) = - \sum_{a,b \in G} \ln \frac{\exp(\text{sim}(\mathbf{h}_a, \tilde{\mathbf{m}}_{a,k})/\tau)}{\sum_{b=1}^{S} \exp(\text{sim}(\mathbf{h}_a, \tilde{\mathbf{m}}_{b,k})/\tau)}, \tag{4}$$

where $\mathbf{h}_a$ is the embedding of instance $a$. $S = |\mathcal{V}|$ or $|\mathcal{E}|$ is the number of instances in $G$, depending on the type of information. The temperature parameter $\tau$ is used to modulate the sharpness of the cosine similarity function $\text{sim}(\cdot)$. Thus, the overall molecular alignment loss is:

$$\mathcal{L}_{\text{align}}(\Phi) = \sum_{k=1}^{K} L_{\text{align}}^k(\phi_k). \tag{5}$$

## 4.3 MOLECULAR ADAPTATION

**Instance-specific token generation.** The aligned 2D topological and molecular knowledge representations are then used for downstream adaptation. Since submolecular instances, including atoms and bonds, manifest distinct topological, physical and chemical characteristics, it is advantageous to adapt to fine-grained, instance-level variations. To this end, for each type of domain knowledge, we employ a conditional network (Zhou et al., 2022; Yu et al., 2025a;b) that conditions on the fused 2D and molecular knowledge representations, and dynamically generates atom- or bond-specific tokens to guide the downstream instance encoding process in an instance-aware manner, thus effectively adapting the pre-trained 2D graph encoder to downstream tasks without updating the entire encoder. Formally, for the $k$-th type of knowledge, we train a conditional network $\text{CondNet}_k$ parameterized

by $\gamma_k$, which generates a set of tokens as follows:

$$\mathbf{T}_k = \texttt{CondNet}_k(\text{Concat}(\mathbf{H}, \tilde{\mathbf{M}}_k); \gamma_k), \tag{6}$$

where $\mathbf{H}$ stands for the atom embeddings $\mathbf{H}_{\mathcal{V}}$ or bond embeddings $\mathbf{H}_{\mathcal{E}}$ depending on the type of knowledge. The conditional network acts as a hypernetwork (Ha et al., 2022)—a compact neural network, such as a multi-layer perceptron (MLP)—that generates instance-specific tokens for the primary network. $\mathbf{T}_k$ share the same dimension with the instance embeddings $\mathbf{H}$. Each row of $\mathbf{T}_k$, denoted $\mathbf{t}_{i,k}$, is a unique adaptation token for the $i$-th instance.

**Instance adaptation.** The generated tokens are then used to guide the adaptation of the pre-trained 2D graph encoder, enabling effective integration with molecular domain knowledge for downstream tasks. On one hand, for atom-level knowledge, atom-specific tokens modulate atom features as

$$\tilde{\mathbf{X}}_k = \mathbf{X} \odot \mathbf{T}_k, \tag{7}$$

where $\mathbf{X}$ is the original atom feature matrix, $\mathbf{T}_k$ is the token matrix for the $k$-th type of knowledge (for atoms), and $\odot$ represents element-wise multiplication. The modified features $\tilde{\mathbf{X}}_k$ are then fed into the pre-trained graph encoder, obtaining the output embedding $\tilde{\mathbf{H}}_k$ through the frozen 2D graph encoder (Eq. 1). On the other hand, for bond-level knowledge, we inject the bond-specific tokens into the message-passing process. Specifically, the $l$-th layer message passing is adapted as

$$\mathbf{h}_{v,k}^l = \texttt{MP}\left(\mathbf{h}_{v,k}^{l-1}, \left\{\mathbf{t}_{(u,v),k} \cdot \mathbf{h}_{u,k}^{l-1} \mid u \in \mathcal{N}_v\right\}; \theta^l\right), \forall v \in \mathcal{V}. \tag{8}$$

We then aggregate the $K$ output embeddings conditioned on the $K$ types of molecular domain knowledge, as follows.

$$\tilde{\mathbf{H}}_{\mathcal{V}} = \sum_{k=1}^{K} \tilde{\mathbf{H}}_{\mathcal{V},k}. \tag{9}$$

Finally, we readout the representation of a molecular graph $G$ by sum pooling:

$$\mathbf{H}_G = \texttt{Readout}(\tilde{\mathbf{H}}_{\mathcal{V}}). \tag{10}$$

**Optimization.** We focus on two popular downstream tasks, namely, molecular property prediction and molecular classification. Given a labeled dataset $\mathcal{D}_{\text{down}} = \{(G_1, y_1), (G_2, y_2), \dots\}$ for a downstream task, where $G_i$ denotes a molecular graph and $y_i$ is its associated label (i.e., a real-valued property or class label), we train a task head to infer the label of a given graph $G$. The task loss is denoted as $\mathcal{L}_{\text{task}}(\Phi, \Gamma, \eta)$, where $\eta$ represent the learnable parameters in the task head, while $\Phi$ and $\Gamma$ represent the parameters of the projectors and conditional networks, respectively. We integrate this task loss with the contrastive alignment loss (Eq. 5) to form the final optimization objective:

$$\mathcal{L}(\Phi, \Gamma, \eta, \beta) = \texttt{Bal}(\mathcal{L}_{\text{task}}(\Phi, \Gamma, \eta), \mathcal{L}_{\text{align}}(\Phi); \beta), \tag{11}$$

where $\texttt{Bal}(\cdot)$ is a balancing function parameterized by $\beta$. In our experiments, we adopt Grad-Norm (Chen et al., 2018) to adaptively balance the contributions of the two objectives. During downstream adaptation, only $\Phi$ in the projectors, $\Gamma$ in the conditional networks, $\eta$ in the task head and $\beta$ in the balancing function are updated, whereas $\Theta$ in the pre-trained encoder are kept frozen. This parameter-efficient design ensures strong performance even under low-resource settings, where the downstream training set $\mathcal{D}$ contains only a few labeled examples. We outline the algorithm of MOLGA in Appendix A.

## 5 EXPERIMENTS

In this section, we conduct experiments to evaluate MOLGA and analyze the empirical results.

### 5.1 EXPERIMENTAL SETUP

**Datasets.** We utilize eleven benchmark datasets for evaluation, with *BBBP* (Martins et al., 2012), *SIDER* (Kuhn et al., 2016), *ClinTox* (Gayvert et al., 2016), *MUV* (Rohrer & Baumann, 2009) and *BACE* (Wu et al., 2018) for molecular classification task, *QM8* (Ramakrishnan et al., 2015), *QM9* (Ramakrishnan et al., 2014), *ESOL* (Delaney, 2004), *Lipophilicity* (Gaulton et al., 2012), *FreeSolv*

Table 1: Summary of the datasets.

| Dataset | Task | No. molecules | Avg. atoms | Avg. bonds | Molecular domain knowledge |
|---|---|---|---|---|---|
| BBBP | Classification | 2,039 | 24.1 | 26.0 | Chemical bonds |
| SIDER | Classification | 1,427 | 33.6 | 35.4 | Chemical bonds |
| ClinTox | Classification | 1,478 | 26.2 | 27.9 | Chemical bonds |
| MUV | Classification | 93,087 | 24.2 | 26.3 | Chemical bonds |
| BACE | Classification | 1,513 | 34.1 | 36.9 | Chemical bonds |
| QM8 | Property prediction | 21,786 | 16.1 | 16.3 | 3D conformation |
| QM9 | Property prediction | 133,885 | 18.0 | 18.4 | 3D conformation |
| ESOL | Property prediction | 1,128 | 13.3 | 13.7 | Chemical bonds |
| Lipophilicity | Property prediction | 4,200 | 27.0 | 29.5 | Chemical bonds |
| FreeSolv | Property prediction | 643 | 8.7 | 8.4 | Chemical bonds |
| MD17-aspirin | Property prediction | 211,762 | 21 | 140.4 | 3D conformation + atom forces |

(Mobley & Guthrie, 2014) and *MD17-aspirin* (Chmiela et al., 2017) for molecular property prediction task. We summarize these datasets in Table 1, and provide detailed descriptions in Appendix B.

**Baselines.** We compare MOLGA with four types of state-of-the-art methods: (1) Supervised 2D graph encoders: GCN (Kipf & Welling, 2017) and GAT (Veličković et al., 2018) are trained directly on downstream labeled data in a fully supervised manner, without any pre-training or use of molecular domain knowledge. (2) Supervised molecular graph encoders: DIMENET++ (Gasteiger et al., 2020), SPHERENET (Liu et al., 2022d), and COMENET (Wang et al., 2022) incorporate molecular domain knowledge into the message passing during downstream training, but without pre-training. (3) 2D graph pretraining method: GRAPHCL (You et al., 2020) and GRAPHPROMPT (Liu et al., 2023c) conduct unsupervised pre-training on unlabeled 2D graphs. They are later adapted to downstream tasks via fine-tuning or prompt learning, while keeping the pre-trained encoder frozen. (4) Molecular graph pretraining methods: GRAPHMVP (Liu et al., 2022a), GEM (Fang et al., 2022), and MOLEBLEND (Yu et al., 2024a) leverage molecular domain knowledge during both the pre-training and adaptation phases. They design self-supervised pre-training objectives grounded in molecular domain knowledge to learn molecular graph encoders, which are subsequently fine-tuned in a uniform manner across all instances for downstream tasks. Detailed descriptions of the baseline methods are provided in Appendix C, with additional implementation details for both the baselines and MOLGA in Appendix D.

**Pre-training setting.** For all baselines and MOLGA, we randomly sample 20% of molecules from the *QM9* dataset (Ramakrishnan et al., 2014), resulting in a subset of 25,000 molecules. Specifically, for 2D graph pre-training methods and MOLGA, only the 2D molecular graphs—without any molecular domain knowledge—are used for pre-training. To further evaluate the effectiveness of MOLGA, we also compare it with molecular graph pre-training methods, which leverage both the 2D graph structure and the associated molecular domain knowledge during pre-training and adaptation. We leverage the their officially released pre-trained molecular graph encoders, which are trained on significantly larger datasets: GRAPHMVP is pre-trained on GEOM (Axelrod & Gomez-Bombarelli, 2022) with 50,000 molecules, GEM on ZINC15 (Sterling & Irwin, 2015) with 2,000,000 molecules, and MOLEBLEND on PCQM4Mv2 (Hu et al., 2021) with 3,370,000 molecules. Since molecular graph pre-training methods utilize substantially more data and incorporate molecular domain knowledge during pre-training, they are not directly comparable to MOLGA.

**Adaptation setting.** We evaluate MOLGA on both molecular classification and molecular property prediction tasks. For molecular classification, we follow an $m$-shot classification setting: for each class, $m$ labeled instances are randomly selected for training, and the remaining instances are used for testing. In our main experiments, we set $m = 5$, which corresponds to less than 1% of the dataset. For molecular property prediction, we randomly sample 100 molecules for downstream adaptation training, 100 for validation, and use the remaining molecules for testing. For *QM9* which is used for pre-training, the samples are drawn from the remaining 80% of data not used during pre-training. For other datasets, sampling is performed from the full dataset.

**Evaluation.** For molecular classification, we follow previous study (Ruddigkeit et al., 2012), employing the ROC-AUC (Bradley, 1997) as the evaluation metric. We construct 100 independent $m$-shot tasks via repeated sampling, and evaluate each task using five distinct random seeds, result-

Table 2: ROC-AUC (%) evaluation of molecular classification.

| Methods | Pre-train | Adapt | BBBP | SIDER | ClinTox | MUV | BACE |
|---|---|---|---|---|---|---|---|
| GCN | × | 2D | 66.23±13.83 | 52.43± 5.30 | 54.59± 5.33 | 52.48± 9.91 | 50.53± 6.20 |
| GAT | × | 2D | 59.72±12.42 | 50.84± 5.50 | 52.91± 4.57 | 51.43±10.41 | 50.21± 5.55 |
| DIMENET++ | × | 2D+Mol | 66.41±10.43 | 50.85± 4.83 | 54.25±10.54 | 54.54± 7.66 | 54.06±13.00 |
| SPHERENET | × | 2D+Mol | 65.54±10.51 | 50.26± 5.92 | 53.83±10.87 | 55.40± 5.79 | 54.43±13.41 |
| COMENET | × | 2D+Mol | 66.89± 9.94 | 50.52± 6.22 | 55.89± 9.09 | 56.69± 6.21 | 54.57±10.21 |
| GRAPHCL | 2D | 2D | 64.61±12.24 | 51.59± 4.45 | 55.58± 6.84 | 58.35± 9.03 | 52.20± 6.19 |
| GRAPHPROMPT | 2D | 2D | 68.88±11.03 | 51.53± 4.20 | 55.67± 7.08 | 59.87± 6.27 | 52.43± 3.62 |
| MOLGA | 2D | 2D+Mol | **71.97**±12.34 | **52.61**± 5.42 | **57.31**± 6.87 | **62.72**± 5.44 | **54.94**± 5.37 |
| Molecular graph pre-training approaches (for reference only) | | | | | | | |
| GRAPHMVP | 2D+Mol | 2D+Mol | 69.05± 8.00 | 50.82± 4.19 | 57.43± 6.12 | 59.59± 7.75 | 55.46± 6.10 |
| GEM | 2D+Mol | 2D+Mol | 71.78± 7.80 | 54.17± 6.30 | 62.32± 7.31 | 53.21± 8.03 | 58.07± 9.24 |
| MOLEBLEND | 2D+Mol | 2D+Mol | 71.45± 9.45 | 52.85± 4.25 | 60.56± 6.36 | 61.81± 7.96 | 57.17± 8.19 |

"×" indicates without pre-training; "2D" denotes the use of 2D topology; "Mol" signifies that molecular domain knowledge is leveraged. Among the comparable methods (excluding those for reference only), the best results are **bolded** and the second-best results are underlined.

Table 3: RMSE evaluation of molecular property prediction.

| Methods | Pre-train | Adapt | QM8 | QM9 | ESOL | Lipophilicity | FreeSolv | MD17-aspirin |
|---|---|---|---|---|---|---|---|---|
| GCN | × | 2D | 0.118±0.003 | 1.400±0.023 | 2.764±0.122 | 1.466±0.131 | 4.298±0.143 | 4.872±0.061 |
| GAT | × | 2D | 0.120±0.003 | 1.160±0.020 | 2.635±0.116 | 1.445±0.068 | 4.267±0.203 | 4.958±0.022 |
| DIMENET++ | × | 2D+Mol | 0.104±0.004 | 1.034±0.054 | 2.377±0.078 | 1.379±0.133 | 4.049±0.213 | 4.781±0.092 |
| SPHERENET | × | 2D+Mol | 0.072±0.007 | 0.928±0.023 | 2.135±0.151 | 1.335±0.127 | 3.900±0.138 | 4.772±0.066 |
| COMENET | × | 2D+Mol | 0.060±0.013 | 0.918±0.042 | 2.317±0.210 | 1.309±0.048 | 4.049±0.285 | 4.755±0.081 |
| GRAPHCL | 2D | 2D | 0.103±0.007 | 1.018±0.011 | 2.038±0.047 | 1.249±0.014 | 4.612±0.133 | 4.900±0.145 |
| GRAPHPROMPT | 2D | 2D | 0.100±0.096 | 0.976±0.022 | 2.052±0.045 | 1.242±0.017 | 4.517±0.129 | 4.843±0.068 |
| MOLGA | 2D | 2D+Mol | **0.056**±0.001 | **0.855**±0.027 | **1.973**±0.043 | **1.187**±0.015 | **3.602**±0.131 | **4.755**±0.049 |
| Molecular graph pre-training approaches (for reference only) | | | | | | | | |
| GRAPHMVP | 2D+Mol | 2D+Mol | 0.032±0.004 | 0.885±0.023 | 2.257±0.043 | 0.945±0.014 | 2.767±0.122 | 4.782±0.033 |
| GEM | 2D+Mol | 2D+Mol | 0.028±0.011 | 0.652±0.003 | 1.175±0.020 | 0.840±0.220 | 1.957±0.073 | 4.761±0.014 |
| MOLEBLEND | 2D+Mol | 2D+Mol | 0.030±0.013 | 0.590±0.009 | 1.496±0.032 | 0.774±0.012 | 1.851±0.068 | 4.708±0.082 |

ing in a total of 500 results. For molecular property prediction, we adopt RMSE (Hodson, 2022) as the evaluation metric, following prior work (Ruddigkeit et al., 2012) and assess with 5 different random seeds. For both molecular classification and property prediction, we report the mean and standard deviation across all runs.

## 5.2 PERFORMANCE EVALUATION

We first evaluate MOLGA on both molecular classification and molecular property prediction tasks. Then we access with various amounts of labeled data for downstream adaptation.

**Molecular classification and property prediction.** We present the results of molecular classification in Table 2 and molecular property prediction in Table 3. We observe that: (1) MOLGA consistently outperforms other baselines, demonstrating the effectiveness of incorporating molecular domain knowledge during the adaptation phase of 2D pre-trained graph encoders; (2) Since molecular graph pre-training methods utilize a *significant larger ($2\times \sim 134\times$) amount of data* for pre-training and are thus not directly comparable, their results are reported only for reference. Nevertheless, MOLGA still achieves competitive performance, further validating its efficacy.

**Various amount of labeled data.** To assess the impact of labeled data size, we evaluate MOLGA and several strong baselines on molecular classification tasks under with various number of shots, as shown in Fig. 3. We make the following observations. (1) MOLGA generally surpasses all competing baselines from 1 to 10 shots, demonstrating its robustness. (2) As the number of labeled instances increases (e.g., $m > 5$), performance typically improves for all models. However, MOLGA exhibits

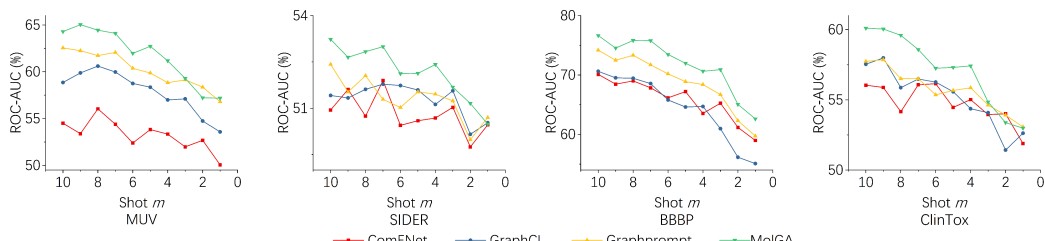

Figure 3: Impact of labeled data size (number of shots) on molecular classification.

Table 4: Ablation study on the effects of different components.

| Methods | AL | CN 2D | CN Mol | BBBP | ClinTox | MUV | BACE | QM8 | ESOL | Lipophilicity | FreeSolv |
|---|---|---|---|---|---|---|---|---|---|---|---|
| | | | | Molecular classification | | | | Molecular property prediction | | | |
| VARIANT 1 | × | × | × | $65.29_{\pm12.04}$ | $55.88_{\pm7.35}$ | $62.61_{\pm4.77}$ | $52.61_{\pm3.33}$ | $0.070_{\pm0.001}$ | $2.035_{\pm0.075}$ | $1.211_{\pm0.023}$ | $3.710_{\pm0.122}$ |
| VARIANT 2 | × | ✓ | × | $70.16_{\pm11.59}$ | $56.66_{\pm6.83}$ | $60.47_{\pm6.48}$ | $53.45_{\pm5.40}$ | $0.057_{\pm0.001}$ | $2.063_{\pm0.042}$ | $1.199_{\pm0.015}$ | $3.716_{\pm0.124}$ |
| VARIANT 3 | × | × | ✓ | $69.87_{\pm13.00}$ | $54.83_{\pm6.69}$ | $59.05_{\pm5.89}$ | $49.96_{\pm2.61}$ | $0.057_{\pm0.001}$ | $2.073_{\pm0.057}$ | $1.200_{\pm0.015}$ | $3.668_{\pm0.097}$ |
| VARIANT 4 | × | ✓ | ✓ | $71.10_{\pm10.71}$ | $56.75_{\pm6.78}$ | $61.40_{\pm6.33}$ | $53.91_{\pm5.98}$ | $0.059_{\pm0.004}$ | $2.018_{\pm0.055}$ | $1.186_{\pm0.010}$ | $3.624_{\pm0.208}$ |
| VARIANT 5 | ✓ | × | × | $71.53_{\pm0.11}$ | $57.13_{\pm6.02}$ | $60.51_{\pm7.11}$ | $54.11_{\pm5.86}$ | $0.059_{\pm0.001}$ | $1.980_{\pm0.042}$ | $1.193_{\pm0.011}$ | $3.610_{\pm0.144}$ |
| MOLGA | ✓ | ✓ | ✓ | $\mathbf{71.97}_{\pm12.34}$ | $\mathbf{57.31}_{\pm6.87}$ | $\mathbf{62.72}_{\pm5.44}$ | $\mathbf{54.94}_{\pm5.37}$ | $\mathbf{0.056}_{\pm0.001}$ | $\mathbf{1.973}_{\pm0.043}$ | $\mathbf{1.187}_{\pm0.015}$ | $\mathbf{3.602}_{\pm0.131}$ |

"AL" is short for molecular alignment; "CN" is short for conditional networks; "2D" denotes conditioning on pre-trained 2D topological knowledge; "Mol" stands for conditioning on molecular domain knowledge.

a steeper performance gain compare to other methods. (3) In low-shot scenarios (e.g., $m \leq 5$), MOLGA remains highly competitive, frequently achieving the best or near-best results.

## 5.3 ABLATION STUDY

To understand the the effect of integration with molecular domain knowledge, molecular alignment and molecular adaptation, we compare MOLGA with five ablated variants and report the results in Table 4. We observe that: (1) Integrating molecular domain knowledge benefits the adaptation of pre-trained 2D graph encoders. This is evidenced by Variant 4 outperforming Variant 2. However, molecular knowledge alone is insufficient, as Variant 4 also surpasses Variant 3, suggesting the necessity of effective integration of both 2D topological and molecular domain knowledge. (2) Molecular alignment effectively bridges the representational gap between pre-trained 2D topological embeddings and associated molecular knowledge, as Variant 5 outperforms Variant 1, and MOLGA further outperforms Variant 4. This highlights the advantage of aligning these two modalities. (3) Conditional networks are beneficial for both molecular classification and molecular property prediction, as MOLGA surpasses Variant 5, and Variant 2 outperforms Variant 1. This demonstrates that modeling instance-specific characteristics—rather than applying a uniform task head—is crucial for fully leveraging both 2D topological structures and molecular domain knowledge.

## 5.4 ADDITIONAL EXPERIMENTS

We conduct additional experiments including: (1) an evaluation of the impact of hyperparameters in Appendix E.1, (2) an analysis of parameter efficiency in Appendix E.2, (3) a study on the flexibility of incorporating various 2D graph pre-training methods in Appendix E.3, and (4) a visualization of MOLGA and its ablated variants in Appendix E.4.

## 6 CONCLUSIONS

In this paper, we studied the adaptation of pre-trained 2D graph encoder to downstream molecular applications by flexibly integrating diverse molecular domain knowledge. Our proposed method, MOLGA, employs a molecular alignment strategy to bridge the gap between the pre-trained 2D topological representation and downstream molecular domain knowledge. Furthermore, we propose a conditional molecular adaptation mechanism that modulates the pre-trained encoder conditioned on aligned topological and molecular knowledge. Extensive experiments on eleven benchmarks demonstrate the effectiveness of MOLGA.

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

# APPENDICES

## A   ALGORITHM

We detailed the key steps for downstream tuning of MOLGA in Algorithm 1. First, we extract molecular domain knowledge and project them into the topological embedding space (line 5–9). Then, in line 10 – 11, we calculate contrastive alignment loss. In line 14–17, we employ conditional networks to generate molecular tokens, which are then used to modify molecular graphs in line 18–24. Specifically, for atom associated molecular domain knowledge, tokens modify the input atom features before fed into the frozen 2D graph encoder (line 19–21 ), while for bond associated molecular domain knowledge, generated tokens are injected into the message-passing process (line 22–24) . Then we readout the molecular embedding and calculate task loss (line 25–28). Finally, we calculate and backpropagate the overall loss and optimize $\Phi, \Gamma, \eta, \beta$ (line 29–30).

## B   FURTHER DESCRIPTIONS OF DATASETS

We summarize all datasets in Table 1 and provide further comprehensive descriptions of these datasets.

- *BBBP* (Martins et al., 2012) is a molecular dataset of 2,039 molecules, each representing a drug-like compound. Atom features denote element types, and labels indicate blood–brain barrier permeability (penetrating or non-penetrating).

---

**Algorithm 1** DOWNSTREAM ADAPTATION FOR MOLGA

---

**Input:** Pre-trained 2D graph encoder GE with frozen parameters $\Theta_0$, labeled data $\mathcal{D}$ for downstream adaptation;
**Output:** Optimized parameters $\Gamma$ in conditional networks, $\Phi$ in projectors, $\eta$ in task head, and $\beta$ in balancing function.

1: $\Phi, \Gamma, \eta, \beta \leftarrow$ initialization
2: **while** not converged **do**
3: $\quad (\mathbf{H}_\mathcal{V}, \mathbf{H}_\mathcal{E}) \leftarrow \texttt{GE}(\mathbf{X}, G; \Theta_0)$
4: $\quad$ /* Molecular alignment */
5: $\quad$ **for** $k = 1$ to $K$ **do**
6: $\quad\quad$ /* Molecular domain knowledge extraction by Eq. 2 */
7: $\quad\quad \mathbf{M}_k \leftarrow \texttt{Extr}_k(M_k)$
8: $\quad\quad$ /* Molecular domain knowledge projection by Eq. 3 */
9: $\quad\quad \tilde{\mathbf{M}}_k \leftarrow f_k(\mathbf{M}_k; \phi_k)$
10: $\quad$ /* Contrastive alignment loss */
11: $\quad$ **for** $k = 1$ to $K$ **do**
12: $\quad\quad$ Calculate $L_{\text{align}}^k(\phi_k)$ by Eq. 4
13: $\quad \mathcal{L}_{\text{align}} \leftarrow \sum_{k=1}^{K} L_{\text{align}}^k$ (Eq. 5)
14: $\quad$ /* Molecular adaptation */
15: $\quad$ /* Molecular token generation by Eq. 6 */
16: $\quad$ **for** $k = 1$ to $K$ **do**
17: $\quad\quad \mathbf{T}_k \leftarrow \texttt{CondNet}_k\big(\text{Concat}(\mathbf{H}, \tilde{\mathbf{M}}_k); \gamma_k\big)$
18: $\quad$ **for** $k = 1$ to $K$ **do**
19: $\quad\quad$ **if** $M_k$ is atom-level **then**
20: $\quad\quad\quad \tilde{\mathbf{X}}_k \leftarrow \mathbf{X} \odot \mathbf{T}_k$
21: $\quad\quad\quad \tilde{\mathbf{H}}_{\mathcal{V},k} \leftarrow \texttt{GE}(\tilde{\mathbf{X}}_k, G; \Theta)$ (Eq. 7)
22: $\quad\quad$ **else if** $M_k$ is bond-level **then**
23: $\quad\quad\quad$ **for** Each layer $l$ in GE **do**
24: $\quad\quad\quad\quad \mathbf{h}_{v,k}^l = \texttt{MP}\left(\mathbf{h}_{v,k}^{l-1}, \left\{\mathbf{t}_{(u,v),k} \cdot \mathbf{h}_{u,k}^{l-1} \mid u \in \mathcal{N}_v\right\}; \theta^l\right), \forall v \in \mathcal{V}$ (Eq. 8)
25: $\quad \tilde{\mathbf{H}}_\mathcal{V} \leftarrow \sum_{k=1}^{K} \tilde{\mathbf{H}}_{\mathcal{V},k}; \quad \mathbf{H}_G \leftarrow \texttt{Readout}(\tilde{\mathbf{H}}_\mathcal{V})$
26: $\quad$ /* Task loss */
27: $\quad \hat{y} \leftarrow \texttt{TaskHead}(\mathbf{H}_G; \eta); \quad \mathcal{L}_{\text{task}} \leftarrow L(\hat{y}, y)$
28: $\quad$ Calculate $\mathcal{L}(\Phi, \Gamma, \eta, \beta)$ by Eq. 11
29: $\quad$ Update $\Phi, \Gamma, \eta, \beta$ by backpropagating $\mathcal{L}(\Phi, \Gamma, \eta, \beta)$
30: **return** $\{\Phi, \Gamma, \eta, \beta\}$

---

- *SIDER* (Kuhn et al., 2016) contains 1,427 molecules annotated with adverse drug reaction categories. Labels represent the presence or absence of hepatobiliary adverse effects.

- *ClinTox* (Gayvert et al., 2016) includes 1,478 molecules associated with toxicity-related information. Labels denote FDA approval status (approved or not approved).

- *MUV* (Rohrer & Baumann, 2009) comprises 93,087 molecules curated for virtual screening across 17 assays. Labels indicate activity against Sphingosine-1-Phosphate Receptor 1 (active or inactive).

- *BACE* (Wu et al., 2018) is a set of 1,513 molecules targeting $\beta$-secretase 1 (BACE-1) inhibition. Labels denote active or inactive status of the molecule, derived from $IC_{50}$ measurements.

- *QM8* (Ramakrishnan et al., 2015) consists of 21,786 molecules with associated excited-state properties, including excitation energies and oscillator strengths.

- *QM9* (Ramakrishnan et al., 2014) contains 133,885 small organic molecules with multiple quantum-chemical properties (e.g., $U_0$, enthalpy, free energy, heat capacity, dipole, polarizability, HOMO/LUMO energies, and energy gap). Each molecule includes both atom/bond features and equilibrium 3D coordinates.

- *ESOL* (Delaney, 2004) is a curated dataset of 1,128 molecules aimed at predicting aqueous solubility, a key pharmacokinetic property.

- *Lipophilicity* (Gaulton et al., 2012) includes 4,200 molecules for predicting the octanol/water distribution coefficient, an important descriptor of bioavailability and membrane permeability.

- *FreeSolv* (Mobley & Guthrie, 2014) provides 642 molecules with experimentally measured hydration free energies ($\Delta G$). The dataset targets subtle intermolecular effects and is used for regression tasks involving noncovalent interactions.

- *MD17* (Chmiela et al., 2017) contains 211,762 conformations of $C_9H_8O_4$ sampled at different time steps, each associated with varying levels of molecular energy.

## C   FURTHER DESCRIPTIONS OF BASELINES

In this section, we provide additional details about the baselines used in our experiments.

(1) Supervised 2D graph neural networks.

- **GCN** (Kipf & Welling, 2017): A graph neural network that aggregates node information using mean-pooling, thereby enabling nodes to capture structural information from their neighbors.

- **GAT** (Veličković et al., 2018): Unlike GCN, GAT incorporates attention mechanisms to assign different weights to neighboring nodes, refining the aggregation process based on their relative importance.

(2) Supervised molecular graph encoders.

- **DimeNet++** (Gasteiger et al., 2020): A directional message-passing network for molecular graphs that encodes inter-atomic distances and angles using spherical Bessel functions and spherical harmonics, enabling efficient modeling of three-body interactions and improved stability over DimeNet (Gasteiger et al., 2021).

- **SphereNet** (Liu et al., 2022d): A spherical message-passing architecture that represents local geometry in spherical coordinates and explicitly incorporates distance, bond angle, and dihedral angle terms, strengthening the capture of 3D structural dependencies.

- **ComENet** (Wang et al., 2022): A complete-and-efficient molecular graph encoder that augments standard edge-to-node updates with angle/dihedral-aware interactions, allowing simultaneous modeling of *2-body*, *3-body*, and *4-body* geometric relations while maintaining computational efficiency.

(3) 2D graph pre-training methods.

- **GraphCL** (You et al., 2020): A contrastive pre-training framework that maximizes agreement between two stochastically augmented views of the same graph via an InfoNCE objective. It employs graph-specific augmentations and an MLP projection head; negatives are other graphs in the batch. This yields transferable 2D topological representations under minimal task-specific supervision.
- **GraphPrompt** (Liu et al., 2023c): A prompt-based adaptation method for graphs in which small, learnable prompt vectors are injected into the input or intermediate layers to steer a (typically frozen) encoder toward downstream tasks with minimal parameter updates.

(4) Molecular graph pre-training methods.

- **GraphMVP** (Liu et al., 2022a): A multi-view pre-training framework that aligns 2D molecular graphs with 3D conformations via contrastive objectives and complementary self-supervised tasks, encouraging cross-modal consistency between topology and geometry.
- **GEM** (Fang et al., 2022): A geometry-enhanced molecular pretraining approach that couples 2D structural cues with 3D geometric supervision (e.g., distance/angle-aware tasks and masked attribute recovery) to learn spatially informed representations.
- **MoleBlend** (Yu et al., 2024a): A relation-level, blend-then-predict pretraining strategy that first blends 2D/3D atom-relation signals into a unified input for a single encoder, then predicts modality-specific targets, enabling fine-grained 2D–3D alignment within one model.

## D  IMPLEMENTATION DETAILS

**Environment.** The environment in which we run experiments is:

- Operating system: Ubuntu 22.04.1
- CPU information: Intel(R) Xeon(R) Platinum 8368Q
- GPU information: NVIDIA L40(48G)

**Optimizer.** For all experiments, we use the Adam optimizer.

**Details of baselines.** For all open-source baselines, we leverage the officially provided code. Each method is tuned based on the settings recommended in their respective literature to achieve optimal performance.

For both GCN(Kipf & Welling, 2017) and GAT(Veličković et al., 2018), we employ a 3-layer architecture, and set the hidden dimension to 64.

For GraphCL (You et al., 2020), a 3-layer GCN is also employed as its base model, with the hidden dimension set to 64. Specifically, we select edge dropping as the augmentations, with a default augmentation ratio of 0.2.

For GraphPrompt (Liu et al., 2023c), a 3-layer GCN is used as the base model for all datasets, with the hidden dimension set to 64.

For DimeNet++ (Gasteiger et al., 2020), we use the default settings provided in the original implementation: 4 interaction layers, hidden dimension of 128, and a cutoff radius of 5.0.

For SphereNet (Liu et al., 2022d), we adopt the default configuration: 4 layers with a hidden dimension of 128 and cutoff radius of 5.0. The basis sizes are kept as default with 7 spherical harmonics and 6 radial basis functions. Output block settings are also maintained as default, with no additional hyperparameter tuning.

For ComENet (Wang et al., 2022), we employ the composite-energy message passing scheme to efficiently encode complete 3D information within 1-hop neighborhoods. Following the DIG framework defaults, we configure the model with 4 layers, a hidden dimension of 256, middle dimension of 64, and a cutoff radius of 8.0. The basis sizes are set to 3 spherical harmonics and 2 radial basis functions, and output layers remain unchanged.

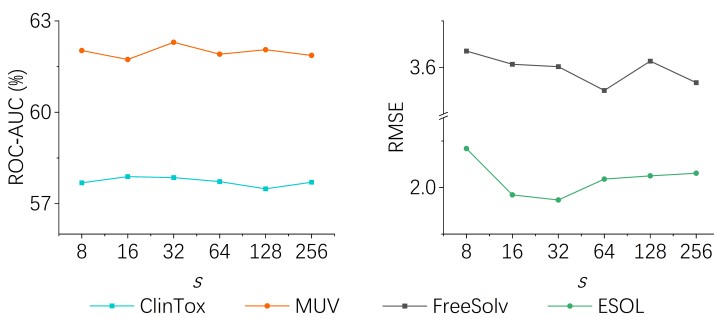

Figure 4: Impact of hidden dimension $s$ in the conditional networks.

Table 5: Comparison of the number of tunable parameters during the downstream adaptation stage.

| Methods | No. parameters |
|---|---|
| GCN | 33,537 |
| GAT | 17,921 |
| DIMENET++ | 1,919,750 |
| SPHERENET | 1,916,870 |
| COMENET | 996,236 |
| GRAPHCL | 128 |
| GRAPHPROMPT | 64 |
| MOLGA | 13,824 |

For GraphMVP (Liu et al., 2022a), We utilize a 5-layer GIN as the 2D encoder with a hidden dimension of 300. For the molecular graph encoder, we adopt a 6-layer SchNet (Liu et al., 2022c) with a hidden dimension of 128.

For GEM (Fang et al., 2022), we employ GeoGNN (Fang et al., 2022) as the molecular graph encoder, consisting of 8 blocks with a hidden dimension of 32.

For MoleBlend (Yu et al., 2024a), we use a unified Transformer backbone with 12 layers and 32 attention heads, where the hidden dimension is set to 768.

**Details of MOLGA.** For our proposed MOLGA, we follow COMENET (Wang et al., 2022) to extract 3D conformations and chemical bond types as bond-level attributes. For atomic force, we directly use the original energy values as atom-level attribute. We using GCN as backbones, which hidden dimension is set to 64. We use a dual-layer MLP with bottleneck structure as the conditional network, and set the hidden dimension of the conditional network as 32.

# E    ADDITIONAL EXPERIMENTS

## E.1    IMPACT OF HYPERPARAMETERS

In our experiments, the conditional network is implemented as a two-layer MLP. To investigate the impact of its design, we vary the hidden dimension $s$ of conditional network and report the results in Fig. 4. We observe that $s = 32$ generally achieves best or near-best performance on both molecular classification and molecular property prediction tasks, thus we set $s = 32$ in our experiments.

## E.2    PARAMETER EFFICIENCY

Finally, we evaluate the parameter efficiency of MOLGA by comparing it with representative baselines. We report the number of parameters that need to be tuned during the downstream adaptation phase, as shown in Table 5. We draw the following observations: (1) Supervised learning methods, including GCN, GAT, DIMENET++, SPHERENET, and COMENET, are trained end-to-end, requir-

Table 6: Evaluation of MOLGA with different 2D graph pre-training methods.

| Pre-train Method | Molecular classification | | | | Molecular property prediction | | | |
|---|---|---|---|---|---|---|---|---|
| | SIDER | ClinTox | MUV | BACE | QM8 | ESOL | Lipophilicity | FreeSolv |
| JOAOv2 | 52.61±5.42 | 57.31±6.87 | 62.72±5.44 | 54.94±5.37 | 0.056±0.001 | 1.878±0.042 | 1.187±0.015 | 3.622±0.131 |
| GraphCL | 51.61±5.33 | 57.74±6.23 | 60.81±6.42 | 54.17±5.71 | 0.059±0.001 | 1.950±0.044 | 1.189±0.015 | 3.530±0.092 |

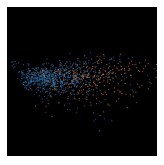 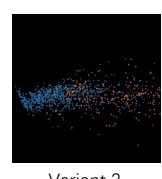 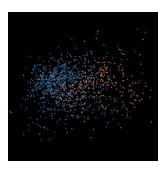 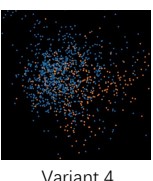 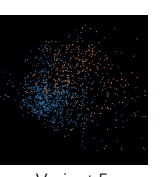 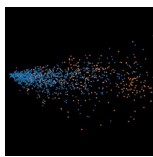

Variant 1     Variant 2     Variant 3     Variant 4     Variant 5     MolGA

Figure 5: Visualization of embedding space of atoms.

ing all model parameters to be updated during downstream training. Thus, these models exhibit poor parameter efficiency. Moreover, compared to GCN and GAT, which only utilize 2D topological structures, the others further incorporate molecular domain knowledge, resulting in a significantly larger number of trainable parameters. (2) 2D graph pre-training-based methods, GRAPHCL, GRAPHPROMPT freeze the pre-trained 2D graph encoder during downstream adaptation, only tune a task head, significantly improving parameter efficiency. (3) MOLGA achieves the best parameter efficiency compared to both supervised learning and molecular graph pre-training methods. Although MOLGA introduces additional lightweight modules (projection heads and conditional networks). While MOLGA tunes slightly more parameters than 2D pre-training methods, the increase is negligible, and does not pose a bottleneck in practice.

### E.3 IMPACT OF PRE-TRAIN METHOD ON DOWNSTREAM PERFORMANCE

To assess the impact of different 2D graph pre-training methods, we employ two representative approaches, GraphCL and JOAOv2 (You et al., 2021), and report the results in Table 6. We observe that MOLGA consistently outperforms comparable baselines across all pre-training methods, demonstrating its flexibility and general applicability. Moreover, using JOAOv2 as the pre-training method generally yields better performance than GraphCL, thus we adopt JOAOv2 in our main experiments.

### E.4 VISUALIZATION

To further demonstrate the effectiveness of our core components, we present the atom embedding space on the *BBBP* dataset in Fig. 5, with different colors indicating distinct atom classes. We observe that, with the incorporation of molecular alignment and conditional adaptation, atom embeddings from different classes exhibit clear separation. This structure reflects a well-organized latent space shaped jointly by 2D topological information and molecular domain knowledge, highlighting the effectiveness of MOLGA.

## F THE USE OF LARGE LANGUAGE MODELS (LLMS)

LLM was used in the preparation of this manuscript exclusively as language-polishing tools. Specifically, we employed LLM to refine grammar, improve clarity, and enhance readability of the text. All research ideas, methodologies, experiments, analyses, and conclusions presented in this work are entirely the authors' own. The LLMs did not contribute to the conception of the research, experimental design, data analysis, or interpretation of results. The authors take full responsibility for the accuracy, originality, and integrity of all scientific content.

# G  DATA ETHICS STATEMENT

We conducted experiments using only publicly available benchmark datasets, i.e., BBBP[1], SIDER[2], ClinTox[3], MUV[4], BACE[5], QM8[6], QM9[7], ESOL[8], Lipophilicity[9], FreeSolv[10], and MD17-aspirin[11]. All datasets were utilized in compliance with their respective terms of use. We further confirm that our research does not involve human participants, animal subjects, or any form of personally identifiable information.

---

[1]http://deepchem.io.s3-website-us-west-1.amazonaws.com/datasets/BBBP.csv

[2]http://deepchem.io.s3-website-us-west-1.amazonaws.com/datasets/sider.csv.gz

[3]http://deepchem.io.s3-website-us-west-1.amazonaws.com/datasets/clintox.csv.gz

[4]https://s3-us-west-1.amazonaws.com/deepchem.io/datasets/muv.csv.gz

[5]http://deepchem.io.s3-website-us-west-1.amazonaws.com/datasets/bace.csv

[6]http://deepchem.io.s3-website-us-west-1.amazonaws.com/datasets/gdb8.tar.gz

[7]http://deepchem.io.s3-website-us-west-1.amazonaws.com/datasets/gdb9.tar.gz

[8]http://deepchem.io.s3-website-us-west-1.amazonaws.com/datasets/delaney-processed.csv

[9]http://deepchem.io.s3-website-us-west-1.amazonaws.com/datasets/Lipophilicity.csv

[10]https://s3-us-west-1.amazonaws.com/deepchem.io/datasets/molnet_publish/FreeSolv.zip

[11]http://quantum-machine.org/gdml/data/npz/md17_aspirin.npz

