# OpenReview forum: "MolGA: Molecular Graph Adaptation with Pre-trained 2D Graph Encoder"
_ICLR.cc/2026/Conference — ICLR 2026 Conference Withdrawn Submission_

### Official Review · Reviewer_Jhnr · 2025-10-26

**Soundness:** 2
**Presentation:** 2
**Contribution:** 2
**Rating:** 4
**Confidence:** 4

**Summary:**

This paper proposes MOLGA, a framework designed to adapt pre-trained 2D graph encoders for molecular learning tasks by integrating diverse molecular domain knowledge. To achieve this, the authors introduce an adaptive fine-tuning strategy that enhances the downstream performance of 2D pre-trained encoders. The main technical contribution is a conditional adaptation mechanism, which enables instance-specific conditioning without re-pretraining large molecular encoders. Specifically, MOLGA aligns 2D topological representations with molecular knowledge via contrastive alignment, followed by lightweight conditional adaptation networks. The approach is parameter-efficient and aims to leverage existing mature 2D pre-trained models while benefiting from molecular-specific domain information.

**Strengths:**

1. The paper identifies a practically relevant and underexplored directio, adapting pre-trained 2D encoders to molecular tasks using domain knowledge rather than conducting new molecular pretraining.
2. The proposed framework is reasonable:
- The contrastive alignment effectively bridges the gap between topological features from pretraining and molecular domain knowledge.
- The conditional adaptation mechanism is efficient and compact, enabling instance-level adaptation without fine-tuning large encoders.

**Weaknesses:**

1.	Readability and logical consistency require improvement.
- In Line 13, the authors state that prior 2D encoders “overlook the rich molecular domain knowledge associated with submolecular instances (atoms and bonds),” yet Line 37 notes that “early studies encode 2D topological structures where nodes represent atoms and edges represent chemical bonds.” These statements appear contradictory.
- In Line 82, the preceding sentence emphasizes the advantages of 2D pretraining, but the following “Moreover, existing…” abruptly shifts to a contrast about “overlooking molecular information.” Lacks coherence.
2.	Limited novelty. Prior works such as GraphMVP already encode atom- and bond-level features in 2D encoders. The authors’ focus on instance-specific chemical or physical variations among similar topological patterns (Line 95) mainly relates to substructure-level variability, which has been extensively investigated [1-2]. The core novelty thus lies in combining molecular graphs with an adaptive mechanism rather than introducing fundamentally new algorithmic designs.
3.	Lack of interpretability analysis. Although atom and bond information are claimed to be incorporated, it remains unclear what molecular patterns the conditional network actually captures. Visualization or case studies would significantly improve interpretability and provide insights into the model’s learned representations.
4.	Insufficient experimental comparison.
- Please add results on Tox21, ToxCast, HIV, and BACE datasets in Table 2.
- The baselines used are relatively outdated; comparisons with recent SOTA models [2-3] are necessary for a convincing evaluation.



[1] Property-Aware Relation Networks for Few-Shot Molecular Property Prediction

[2] Pin-Tuning: Parameter-Efficient In-Context Tuning for Few-Shot Molecular Property Prediction

[3] Uni-Mol: A Universal 3D Molecular Representation Learning Framework

**Questions:**

1. The paper emphasizes “molecular domain knowledge at the submolecular level (atoms and bonds).” How does this differ conceptually from the use of substructures or functional groups in prior works, aside from differences in extraction methodology?
2. What is the training cost or computational overhead of conditional networks compared with standard fine-tuning?
3. Additional questions are embedded in the Weaknesses.

---

### Official Review · Reviewer_7wUa · 2025-10-28

**Soundness:** 2
**Presentation:** 1
**Contribution:** 2
**Rating:** 4
**Confidence:** 5

**Summary:**

This paper introduces MolGA, a molecular graph adaptation framework designed to encode both local atomic environments and global topological structures by incorporating molecular alignment and adaptation to external pretrained models. The motivation is clear and relevant, as molecular property prediction often suffers from limited labeled data, and leveraging pretrained knowledge is a logical direction for improvement. The proposed approach reportedly enhances the utilization of domain-specific information and demonstrates gains across various molecular benchmarks in both regression and classification tasks.

The work’s main strength lies in its clear motivation and the empirical performance improvements observed over conventional, non-pretrained graph models. The idea of combining pretrained molecular knowledge with graph adaptation is conceptually appealing and could have strong implications for multi-task and transfer learning in chemistry.

However, the paper falls short in several key aspects. The description of the initial knowledge embedding method—central to how MolGA captures topological graph information—is missing or incomplete, despite being referenced as provided in the appendix. The dataset section is vague and does not specify which tasks were used, nor does it explain the selection criteria. For instance, QM9 contains multiple tasks such as HOMO, LUMO, Bandgap, etc... yet the specific information is not contained in the script. The ablation studies mention multiple “variants” of MolGA but do not define or distinguish them, making it difficult to interpret the reported results. Furthermore, the paper lacks any computational cost or efficiency analysis, even though the method involves multiple similarity and alignment operations that could increase runtime complexity. Finally, the comparative scope is somewhat limited: since MolGA leverages pretrained models, it should also be evaluated against other pretrained baselines, not only standard non-pretrained graph neural networks.

Overall, MolGA is a promising concept with tangible improvements over simple baselines, but the presentation is incomplete and lacks sufficient methodological and experimental clarity for full evaluation. A stronger exposition of the embedding mechanism, experimental details, ablation structure, and computational profile would significantly improve the paper’s credibility.

**Strengths:**

- The paper addresses an important and timely problem — leveraging pretrained molecular knowledge for better generalization under limited data.

- Demonstrates consistent performance improvement over non-pretrained baseline models.

- The proposed idea of aligning molecular representations with pretrained knowledge is conceptually interesting and worth exploration.

**Weaknesses:**

(1) Lack of clarity and self-containment:
- The manuscript refers to initial knowledge embedding methods and related equations in the appendix, but these details are missing or insufficiently described. This omission makes it difficult to understand how the proposed model captures topological graph information.

(2) Incomplete dataset description:
- The dataset section lacks key information about which tasks or molecular properties were evaluated. For instance, the QM9 dataset includes multiple targets (HOMO, LUMO, Bandgap, etc.), but the specific ones used are not identified.

(3) Unclear ablation study design:
- The paper mentions “variants” of MolGA in its ablation studies but does not define what these variants represent or how they differ from the base model.

(4) Missing computational cost analysis:
- Given that MolGA includes multiple similarity and alignment operations, a computational complexity or runtime comparison would be valuable for understanding its scalability.

(5) Limited comparative scope:
- Since MolGA leverages pretrained representations, it should be compared not only to non-pretrained baselines but also to recent pretrained molecular models. Even if MolGA underperforms on raw accuracy, demonstrating complementary advantages (e.g., lower data requirements, interpretability, transferability) would strengthen the contribution.

**Questions:**

- What is the exact initial knowledge embedding method used in MolGA?

- Which specific tasks or properties were included in the QM9 experiments?

- Can the authors provide a computational cost or efficiency comparison?

- What are the defined “variants” in the ablation studies?

---

### Official Review · Reviewer_QrJo · 2025-10-28

**Soundness:** 3
**Presentation:** 3
**Contribution:** 2
**Rating:** 4
**Confidence:** 4

**Summary:**

The paper presents a framework for adapting pre-trained 2D molecular graph encoders to downstream molecular learning tasks while flexibly integrating domain knowledge. The authors identify that while existing pre-trained 2D graph models excel in topology-based learning, they overlook rich molecular knowledge such as atomic properties or submolecular functional patterns. The author introduces two main innovations to address the problem: (1) molecular alignment strategy which aligns topological embeddings from pre-trained encoders with molecular domain knowledge representations to bridge semantic gaps. (2) conditional adaptation mechanism which generates instance-specific tokens to inject domain knowledge dynamically during fine-tuning.

**Strengths:**

1.	The paper observes a key problem: adapting pre-trained 2D graph encoder to enhance performance for downstream tasks while integrating different types of molecular domain knowledge. To address this, the author proposed a parameter-efficient method by first aligning pre-trained 2D topological representations with molecular domain knowledge and then adapt to downstream tasks. Overall, I think the observation is novel and the proposed method is conceptually neat and potentially extensible.
2.	The paper is generally well-organized and clearly written. The motivation is presented logically, figures clearly communicate the architectural design, and the experimental section is easy to follow.

**Weaknesses:**

1.	The framework seems like that it can serve as a general recipe for incorporating domain priors into pre-trained GNNs. But there is no experiment about other molecular encoders
2.	The baseline methods used in the experiments are somewhat outdated, with most comparisons limited to works from around 2022.

**Questions:**

1.	Could the authors provide additional results or analysis using different molecular encoders (e.g., GIN, GAT, Graphormer)? This would help demonstrate whether MolGA’s effectiveness generalizes beyond the specific encoder tested in the paper.
2.	Could the authors include or discuss results against more recent baselines?
3.	Could the authors further clarify MolGA’s advantages over established 2D molecular graph pre-training methods such as GraphMAE [1], Uni-Mol [2], and SimSGT [3]? The reported results in MolGA appear weaker or less competitive compared to these models. In addition, could the authors report or discuss the computational efficiency of the proposed approach relative to these prior methods?

[1] GraphMAE: Self-Supervised Masked Graph Autoencoders

[2] Uni-Mol: A Universal 3D Molecular Representation Learning Framework

[3] Rethinking Tokenizer and Decoder in Masked Graph Modeling for Molecules

---

### Official Review · Reviewer_P4SE · 2025-10-28

**Soundness:** 3
**Presentation:** 3
**Contribution:** 2
**Rating:** 4
**Confidence:** 4

**Summary:**

MolGA is a framework that adapts pre-trained 2D molecular graph encoders for downstream tasks by flexibly integrating diverse molecular domain knowledge through contrastive alignment and instance-specific conditional adaptation. It keeps the 2D encoder frozen and injects domain knowledge via lightweight conditional networks, enabling fine-grained adaptation without retraining the encoder. Extensive experiments on eleven public datasets demonstrate its effectiveness in molecular classification and property prediction tasks.

**Strengths:**

- Parameter-efficient adaptation: MolGA adapts pre-trained 2D molecular graph encoders to new tasks without re-training them, using a frozen-encoder design and lightweight conditional networks that greatly reduce computational cost and parameters.
- Structured integration of domain knowledge: Through molecular alignment and conditional adaptation, MolGA can incorporate various forms of molecular domain knowledge (e.g., 3D conformation, bond type, atom energy) in a unified and fine-grained manner during downstream adaptation.
- Comprehensive evaluation: MolGA demonstrates strong and stable performance on eleven benchmark datasets for both molecular classification and property prediction, achieving robustness even in low-resource (few-shot) settings.

**Weaknesses:**

- Limited methodological novelty: The key components—contrastive alignment and a frozen encoder with lightweight conditional adaptation—have been previously established in other domains (e.g., CLIP, GraphPrompt).
- Weak justification for the two-stage design: The proposed benefit of first pre-training a 2D encoder and later injecting domain knowledge during downstream adaptation is not convincingly justified. Since the additional knowledge (3D conformation, bond, energy) can be directly extracted from the same dataset, one could feasibly incorporate it within the original molecular pre-training itself.
- Limited empirical support: While MolGA shows improvements over 2D baselines, it still lags behind molecular pre-training methods (e.g., GraphMVP, GEM, MoleBlend), even if listed as “reference only.” This weakens the claim of clear competitiveness.
- Restricted diversity of domain knowledge: The paper claims to incorporate diverse molecular knowledge; however, only three types (3D conformation, chemical bond, and atom energy) are used.

**Questions:**

- Clarification on Eq. (4) Contrastive Alignment: Does the contrastive alignment loss compute pairwise similarities across all atomic/bond instances within each molecule, or only within the mini-batch? The current description (Eq. 4) is ambiguous.
- Effect and necessity of Contrastive Alignment: Ablation results (Table 4, Variant 4 vs. MolGA) show only marginal improvements when alignment is included. Could the authors provide a deeper explanation or analysis of why aligning 2D and molecular knowledge embeddings should theoretically (or empirically) improve downstream performance?
- On extending the proposed mechanism to existing molecular pre-training models: Have the authors considered applying the same alignment and adaptation mechanism on top of molecular pre-trained encoders such as GraphMVP, GEM, or MoleBlend? This could better reveal whether MolGA's contribution is complementary or merely redundant.

---

### Official Review · Reviewer_AjQp · 2025-11-01

**Soundness:** 2
**Presentation:** 2
**Contribution:** 2
**Rating:** 2
**Confidence:** 4

**Summary:**

This paper introduces MolGA, a framework designed to adapt pre-trained 2D graph encoders for downstream molecular tasks. The core idea is to enhance these 2D encoders, which typically only capture topology, by incorporating molecular domain knowledge (e.g., 3D conformations, atom energy) during the adaptation phase. The proposed method keeps the 2D graph encoder frozen, promoting parameter efficiency. It consists of two stages: (1) a "Molecular Alignment" stage, which uses a contrastive loss to train projectors that map domain knowledge representations into the 2D encoder's embedding space, and (2) a "Conditional Adaptation" stage, where lightweight conditional networks generate instance-specific (atom/bond) tokens. These tokens then modulate the inputs and message-passing steps of the frozen 2D encoder to perform downstream tasks. The authors evaluate this approach on several public datasets.

**Strengths:**

- The paper aims at tacklinmg a practical and important problem in molecular representation learning.
- The proposed architecture is explained well.

**Weaknesses:**

- The paper's experiments are confined to relatively small benchmark datasets. To truly validate the method's effectiveness and scalability, it should have been tested on large-scale datasets like PCQM4Mv2 or benchmarks from the Open Catalyst Project.
- It would be better to include a baseline that simply finetunes the 25k pre-trained 2D encoder on the downstream tasks. This shows if the complex MolGA framework provides any actual benefit over standard finetuning.
- The proposed adaptation, as described by Eq. 7, 8, and 9, appears to require a separate forward pass or, at minimum, a separate set of message-passing computations for each of the $K$ types of domain knowledge. This design scales computational cost linearly with the number of knowledge types, $K$. This seems inefficient and a practical limitation.
- Some related work should be cited and discussed:

[1] Dwivedi, V. P., & Bresson, X. (2020). A generalization of transformer networks to graphs. arXiv. arXiv preprint arXiv:2012.09699.

[2] Kreuzer, D., Beaini, D., Hamilton, W., Létourneau, V., & Tossou, P. (2021). Rethinking graph transformers with spectral attention. Advances in Neural Information Processing Systems, 34, 21618-21629.

[3] Kim, J., Nguyen, D., Min, S., Cho, S., Lee, M., Lee, H., & Hong, S. (2022). Pure transformers are powerful graph learners. Advances in Neural Information Processing Systems, 35, 14582-14595.

[4] Rampášek, L., Galkin, M., Dwivedi, V. P., Luu, A. T., Wolf, G., & Beaini, D. (2022). Recipe for a general, powerful, scalable graph transformer. Advances in Neural Information Processing Systems, 35, 14501-14515.

[5] Luo, S., Chen, T., Xu, Y., Zheng, S., Liu, T. Y., Wang, L., & He, D. One Transformer Can Understand Both 2D & 3D Molecular Data. In The Eleventh International Conference on Learning Representations.

**Questions:**

- What was the reasoning for choosing element-wise multiplication for the bond-level adaptation in Eq. 8? Why this specific mechanism?
- Why did you choose to pretrain your own encoder on 25k samples from QM9 instead of using a larger-scale pre-trained GNN?

---

### Official Review · Reviewer_7aTo · 2025-11-02

**Soundness:** 3
**Presentation:** 3
**Contribution:** 2
**Rating:** 2
**Confidence:** 4

**Summary:**

This paper proposes MOLGA, a framework for molecular graph adaptation that reuses a pre-trained 2D graph encoder (e.g., GNN or Graph Transformer) and adapts it to downstream molecular tasks by flexibly incorporating multiple types of molecular domain knowledge (e.g., 3D conformation, chemical bonds, atomic energy). The framework consists of two main modules: (1) molecular alignment and (2) molecular adaptation. Experiments show its improved performance compared with various baselines while using less data. Ablation studies demonstrate contributions from both the alignment and conditional adaptation modules.

**Strengths:**

- The motivation is clear, i.e. how to reuse existing pre-trained 2D graph encoders for molecular tasks requiring richer domain information without retraining large molecular encoders.
- The combination of contrastive alignment (topological vs. molecular representations) and conditional token-based adaptation is technically plausible. The design resembles CLIP-style multimodal alignment coupled with parameter-efficient prompting ideas.
- Keeping the backbone frozen while learning only small projectors and conditional networks makes the method computationally attractive for low-resource regimes.
- The paper is clearly organized, with intuitive figures.

**Weaknesses:**

- While the integration of contrastive alignment and conditional prompting is well performed, both components are adaptations of known techniques (e.g., CLIP alignment, conditional prompts, hypernetworks). The contribution feels more combinatorial than conceptually novel, as no fundamentally new learning principle is introduced for molecular graphs.
- The paper lacks a clear theoretical justification or diagnostic analysis for why contrastive alignment plus conditional tokenization should synergize effectively. The justification is purely empirical.
- The “rule-based extractors” used for generating molecular domain features are not well-explained, e.g. the quality and variety of these features can strongly affect performance, yet the work treats them as a black box.
- Although the authors acknowledge that molecular pre-training baselines use more data, the comparison to 2D-only pre-training is still somewhat uneven. The pre-trained 2D encoder is trained on only a 25k subset of QM9, which might be too small to convincingly demonstrate strong generalization capacity.
- The paper would benefit from more experiments showing generalization to unseen molecular knowledge types or cross-domain transfer.
- Although ablations isolate the contributions of alignment and conditional networks, the analysis remains numerical. Visualization of the aligned embedding spaces or adaptation tokens could better illustrate the claimed representational bridging.
- Given the pace of recent progress in molecular foundation models (e.g., Uni-Mol 2 [1], GEM-2 [2], MolFM [3], etc.), the baselines appear somewhat dated (mostly ≤ 2023). Including or at least discussing comparisons with recent unified 2D-3D or adapter-based molecular pre-training frameworks would considerably strengthen empirical credibility.
- Most benchmarks (BBBP, BACE, etc.) are outdated and too small to stress-test the proposed adaptation mechanism. Evaluating MOLGA on large-scale and more diverse benchmarks, e.g. PCQM4Mv2 [4], TDC [5], or SPICE [6], would better demonstrate scalability and real-world relevance.

[1] Ji, Xiaohong, et al. "Uni-mol2: Exploring molecular pretraining model at scale." arXiv preprint arXiv:2406.14969 (2024).

[2] Liu, Lihang, et al. "GEM-2: next generation molecular property prediction network by modeling full-range many-body interactions." arXiv preprint arXiv:2208.05863 (2022).

[3] Luo, Yizhen, et al. "Molfm: A multimodal molecular foundation model." arXiv preprint arXiv:2307.09484 (2023).

[4] Hu, Weihua, et al. "Ogb-lsc: A large-scale challenge for machine learning on graphs." arXiv preprint arXiv:2103.09430 (2021).

[5] Huang, Kexin, et al. "Therapeutics data commons: Machine learning datasets and tasks for drug discovery and development." arXiv preprint arXiv:2102.09548 (2021).

[6] Eastman, Peter, et al. "Spice, a dataset of drug-like molecules and peptides for training machine learning potentials." Scientific Data 10.1 (2023): 11.

**Questions:**

Minor clarity issues:
- The “Balancing Function” description (Eq. 11) lacks clarity on its dynamics beyond citing GradNorm.

---

### Note · Authors · 2025-12-01

I have read and agree with the venue's withdrawal policy on behalf of myself and my co-authors.